# FFHQ-Makeup: Paired Synthetic Makeup Dataset with Facial Consistency Across Multiple Styles

## Abstract

Paired bare-makeup facial images are essential for a wide range of beauty-related tasks, such as virtual try-on, facial privacy protection, and facial aesthetics analysis. However, collecting high-quality paired makeup datasets remains a significant challenge. Real-world data acquisition is constrained by the difficulty of collecting large-scale paired images, while existing synthetic approaches often suffer from limited realism or inconsistencies between bare and makeup images. Current synthetic methods typically fall into two categories: warping-based transformations and text-to-image generation. The former often distorts facial geometry and compromises makeup precision, while the latter tends to alter facial identity and expression, undermining consistency. In this work, we present *FFHQ-Makeup*, a high-quality synthetic makeup dataset that pairs each identity with multiple makeup styles while preserving facial consistency in both identity and expression. Built upon the diverse FFHQ dataset, our pipeline transfers real-world makeup styles from existing datasets onto 18K identities by introducing an improved makeup transfer method that disentangles identity and makeup. Each identity is paired with 5 different makeup styles, resulting in a total of 90K high-quality bare–makeup image pairs. We release FFHQ-Makeup as the first large-scale, multi-style, paired bare–makeup dataset, which we expect will serve as a valuable resource for future research in beauty-related tasks.

## 1 Introduction

Makeup plays a multifaceted role in human appearance, influencing not only facial aesthetics but also perceptions of identity, personality, and even social behavior. In the context of computer vision, the ability to analyze, manipulate, and transfer makeup styles has drawn increasing attention, enabling applications such as virtual try-on (VTO) Zhang et al. (2024); Alashkar et al. (2017b); Sun et al. (2022); Li et al. (2018a), face recognition under makeup variations Dantcheva et al. (2012); Chen et al. (2013); Sajid et al. (2018), facial privacy protection Hu et al. (2022); Li et al. (2018b), and beauty assessment Xiao et al. (2021). While the field has made remarkable progress in recent years, a critical bottleneck remains: the lack of open-source, large-scale, high-quality paired makeup datasets containing both bare and makeup images. This limitation hinders the development of robust and generalizable models, and significantly impedes progress in makeup-related applications.

A well-constructed paired makeup dataset is expected to meet several key requirements. First, it must maintain high makeup realism, ensuring that the applied styles are both plausible and visually convincing. Second, it should exhibit diversity in both facial identities and makeup styles to reflect real-world variability. Third, it should ensure facial consistency across pairs, preserving identity and facial structure despite the presence of makeup. In addition, the dataset should ideally be large enough to support the data requirements of modern deep learning models.

Despite ongoing efforts, existing makeup datasets construction still fail short of key requirements, as summarized in Tab. 1. Real-world data collection is hindered by logistical constraints, limited resources, and privacy concerns. As a result, these datasets are typically either limited in scale Dantcheva et al. (2012); Chen et al. (2017); Hu et al. (2013) or lack paired bare and makeup images Li et al. (2018a); Gu et al. (2019); Jiang et al. (2020); Nguyen et al. (2021); Yan et al. (2023).

Table 1: **Existing makeup datasets.** Summary of existing makeup datasets, comparing scale, resolution, type, and availability.

| Datasets | Subjects | Images per Subject | Makeup Images | Non-makeup Images | Resolution | Type | Paired | Public avail. |
|---|---|---|---|---|---|---|---|---|
| YMU Dantcheva et al. (2012) | 151 | 4 | 302 | 302 | $130 \times 150$ | Real | ✓ | ✗ |
| MIW Chen et al. (2013) | 125 | 1-2 | 77 | 77 | – | Real | ✗ | ✗ |
| MIFS[1] Chen et al. (2017) | 214 | 2 or 4 | 214 | 428 | – | Real | ✓ | ✗ |
| FAM Hu et al. (2013) | 519 | 2 | 519 | 519 | $64 \times 64$ | Real | ✓ | ✗ |
| MT Li et al. (2018a) | 2719 | 1-2 | 2719 | 1115 | $361 \times 361$ | Real | ✗ | ✓ |
| LADN Gu et al. (2019) | 635 | 1 | 302 | 333 | $\approx 320 \times 320$ | Real | ✗ | ✓ |
| Wild Jiang et al. (2020) | 772 | 1 | 403 | 369 | $256 \times 256$ | Real | ✗ | ✓ |
| CPM-Real Nguyen et al. (2021) | 2895 | 1 | 2895 | – | – | Real | ✗ | ✓ |
| BeautyFace Yan et al. (2023) | 44 | 1+ | 3000 | – | $512 \times 512$ | Real | ✗ | ✓ |
| VMU Dantcheva et al. (2012) | 51 | 4 | 153 | 51 | $130 \times 150$ | Synthetic (Manually edited) | ✓ | ✗ |
| LADN-Syn Gu et al. (2019) | 333 | 355 | 120K | 333 | $\approx 320 \times 320$ | Synthetic (Warp-Paste) | ✓ | ✓ |
| Stable-Makeup Zhang et al. (2024) | 20K | 1 | 20K | 20K | $512 \times 512$ | Synthetic (Text-to-Image[2]) | ✓ | ✗ |
| BeautyBank Lu et al. (2025) | 70K | 1+ | 324K | 70K | $512 \times 512$ | Synthetic (Text-to-Image[2]) | ✓ | ✓ |
| ***FFHQ-Makeup (Ours)*** | 18K | 5 | 90K | 18K | $512 \times 512$ | Synthetic (Generation) | ✓ | ✓ |

[1] For MIFS, makeup images = 214 (imposters only), non-makeup images = 214 (imposters) + 214 (targets) = 428 in total.
[2] Stable-Makeup uses LEDITS Tsaban & Passos (2023) for makeup synthesis, while BeautyBank adopts a same strategy with improved LEDITS++ Brack et al. (2024).

To address these issues, synthetic approaches have recently gained popularity. However, as illustrated in Fig. 1, they still suffer from various limitations: warping-based methods Gu et al. (2019) often leading to artifacts or distortions, while text-to-image generation Zhang et al. (2024); Lu et al. (2025) frequently causes identity drift and inconsistencies in facial expression, and struggles to capture fine-grained makeup details due to the inherent ambiguity of language in describing subtle color gradients, eyeshadow geometry, and cheek contours. Consequently, existing resources offer limited utility for makeup-related applications.

To address these challenges, we present ***FFHQ-Makeup***: an open-source, large-scale, high-quality multiple paired synthetic makeup dataset designed to overcome the limitations of prior work. We introduce a novel data generation method built upon the state-of-the-art makeup transfer method Zhang et al. (2024), with key improvements in both facial structure control and makeup feature extraction. For structure control, our method eliminates the reliance on paired bare-makeup images, which are difficult to obtain in real-world scenarios. Instead, given a single makeup image, we employ a 3D Morphable Model (3DMM) fitting to reconstruct an approximate bare face counterpart.

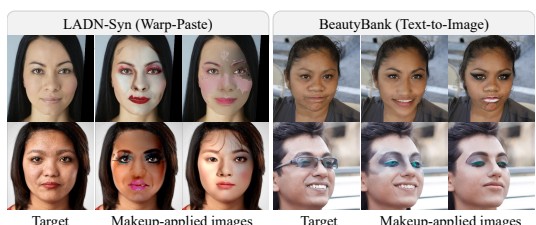

Figure 1: **Examples of existing large-scale synthetic paired bare–makeup datasets.** Existing methods often introduce artifacts or alter the identity and expression of the subject.

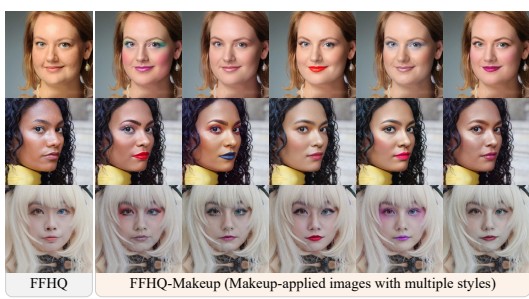

Figure 2: **Examples of FFHQ-Makeup dataset.** Each identity is paired with a bare image and multiple makeup-applied images.

The fitted 3DMM not only captures the subject's identity, expression, pose, and skin tone, but also the illumination. This self-supervised approach enables scalable and flexible for synthetic makeup dataset generation. For makeup feature extraction, we introduce a 3DMM-based residual representation of makeup appearance, combined with sampling and re-rendering augmentation strategies. These techniques extend limited makeup data across diverse facial variations in FFHQ Karras et al. (2019), helping disentangle facial structure from makeup appearance and facilitating the generation of semantically consistent bare–makeup pairs. During dataset construction, we manually filtered out failed cases and samples with visual artifacts to ensure data quality. The final FFHQ-Makeup dataset comprises 18K identities derived from FFHQ, each paired with 5 distinct makeup styles, resulting

in a total of 90K paired images, as illustrated in Fig. 2. We evaluate the effectiveness of both our dataset and method through extensive experiments and comparisons. We anticipate that this high-quality, multi-style paired dataset will greatly benefit a wide range of future makeup-related research and applications.

Our main contributions are as follows:

- **FFHQ-Makeup Dataset:** We introduce a large-scale synthetic dataset with 18K identities and 5 diverse makeup styles per identity, providing 90K high-quality paired bare–makeup images.

- **Pair-free Structure Control:** We propose a scalable generation pipeline that reconstructs bare faces from single makeup images via 3DMM fitting, removing the need for paired supervision.

- **Decoupled Makeup Synthesis:** We extract residual makeup features based on 3DMM, then transfer them across diverse faces using sampling and re-rendering augmentation, effectively disentangling facial structure from makeup appearance.

## 2 RELATED WORKS

### 2.1 MAKEUP DATASETS AND TASKS

We summarize representative makeup-related datasets in Tab. 1. Early studies primarily focused on face recognition and verification tasks under makeup conditions Dantcheva et al. (2012); Chen et al. (2013); Hu et al. (2013); Chen et al. (2017); Sajid et al. (2018), where small-scale, low-resolution datasets were collected to support experimental evaluations. In recent years, the scope of makeup-related research has significantly expanded, including tasks such as facial privacy protection Hu et al. (2022); Li et al. (2018b); Wang et al. (2020), beauty assessment Xiao et al. (2021), makeup recommendation Liu et al. (2022); Alashkar et al. (2017a), and 3D facial makeup Yang & Taketomi (2022); Yang et al. (2023; 2024); Bao et al. (2024). Makeup transfer Li et al. (2018a); Gu et al. (2019); Chen et al. (2019); Jiang et al. (2020); Lyu et al. (2021); Deng et al. (2021); Nguyen et al. (2021); Sun et al. (2022); Yang et al. (2022); Yan et al. (2023); Sun et al. (2024b); Lu et al. (2025); Jin et al. (2024); Sun et al. (2024a); Zhang et al. (2024), in particular, has emerged as the dominant research focus, with deep learning approaches increasingly requiring large-scale, high-quality training data.

Makeup transfer aims to apply the makeup style of a reference image to a target face, while keeping the target's identity, pose, and expression unchanged. Achieving this goal is particularly difficult due to the lack of ground-truth training pairs. Early works adopted GAN-based pipelines. Beauty-GAN Li et al. (2018a) utilized a dual input/output architecture with a color histogram loss to guide region-wise color matching. To train the model, they collected a non-paired dataset (MT) with 2,719 makeup and 1,115 non-makeup images. LADN Gu et al. (2019) introduced multiple local discriminators to better handle heavy makeup styles, collecting 302 makeup and 333 non-makeup images. To augment data, they generated 120K synthetic samples (LADN-Syn) using warp-and-paste, though the realism remains limited. PSGAN Jiang et al. (2020) addressed spatial misalignment via an attention-based makeup projection module, using a dataset of 403 makeup images (mostly side profiles) for evaluation. CPM Nguyen et al. (2021) tackled both color and pattern transfer via UV map representations and synthetic pattern datasets. They also collected a 2,895-image real makeup dataset (CPM-Real). Compared to previous datasets, BeautyREC Yan et al. (2023) increased resolution to $512 \times 512$ and included 3,000 makeup images (BeautyFace).

Recently, diffusion-based approaches have enabled more flexible and fine-grained makeup transfer. Stable-Makeup Zhang et al. (2024) proposed a detail-preserving encoder and incorporated cross-attention layers. They also built a pseudo-parallel dataset of 20K image pairs for training, generated via GPT-4-guided editing Tsaban & Passos (2023) and refined through manual quality filtering. Unfortunately, the dataset is not publicly available. Inspired by Stable-Makeup, BeautyBank Lu et al. (2025) adopted an improved LEDITS++ Brack et al. (2024) strategy to generate a larger dataset. However, due to the lack of post-filtering, the resulting makeup and non-makeup images are often misaligned.

To compensate for the scarcity of high-quality datasets, we propose FFHQ-Makeup to fill a critical gap by offering a publicly available, high-quality, multi-reference paired makeup dataset that supports more robust and scalable research in makeup-related tasks.

## 2.2 FACIAL ATTRIBUTES WITH 3DMM

Since its introduction by Blanz and Vetter Blanz & Vetter (1999), the 3D Morphable Model (3DMM) has been widely used in a variety of face-related tasks Egger et al. (2020). It represents facial shape and texture as linear combinations of basis components learned from a collection of 3D faces. Through 3DMM fitting Zollhöfer et al. (2018), it is possible to recover 3D facial attributes—such as identity, expression, and pose—from a single 2D image. These model-based attributes have proven effective for controllable image synthesis and facial attribute editing, allowing structured manipulation of identity, expression, texture, pose, and illumination Zhao et al. (2023); Jang et al. (2025); Ding et al. (2023); Ponglertnapakorn et al. (2023).

In this work, we leverage the FLAME model Li et al. (2017) to reconstruct a bare face representation from a given makeup image. This enables a synthetic pipeline that no longer relies on paired data, facilitating more flexible dataset generation.

## 2.3 DIFFUSION MODELS FOR FACE EDITING

Diffusion models Ho et al. (2020) are generative models that iteratively transform random noise into realistic images through a sequence of denoising steps. Pretrained latent diffusion models, such as Stable Diffusion Rombach et al. (2022), have demonstrated strong capabilities in photorealistic image synthesis and controllable image editing. Building on this foundation, recent works have extended Stable Diffusion for face-related applications Ye et al. (2023); Wang et al. (2024); Li et al. (2024).

Stable-Makeup Zhang et al. (2024) is the first work to introduce pre-trained Stable Diffusion into the makeup transfer task, setting a new standard in the field. Its architecture consists of two main components: (1) a feature extraction module that employs the CLIP image encoder Radford et al. (2021), aggregating multi-layer features from the visual backbone to capture fine-grained details. Additionally, it incorporates a self-attention-based mapping to more efficiently extract and align makeup features; (2) a structure control module based on ControlNet Zhang et al. (2023), which enables conditional generation guided by specific structural inputs without altering the basic diffusion model. To further improve facial feature extraction, FreeUV Yang et al. (2025) builds on Stable-Makeup by introducing a channel-attention mapping mechanism, which enhances feature discrimination while mitigating the spatial interference introduced by self-attention.

Our method builds upon Stable-Makeup, incorporating the improved feature extraction design from FreeUV to enable robust and reliable makeup dataset generation.

## 3 APPROACH: FFHQ-MAKEUP

We propose that a high-quality paired makeup dataset should satisfy the following three properties:

- **Makeup Realism** ($\mathcal{P}_{makeup}$): The makeup should appear natural and realistic in terms of texture, color, and spatial placement (e.g., lipstick, eyeshadow, blush), closely resembling real-world cosmetic applications.
- **Facial Diversity** ($\mathcal{P}_{diversity}$): The dataset should cover a wide range of facial identities and attributes.
- **Facial Consistency** ($\mathcal{P}_{consistency}$): Each bare-makeup image pair should preserve consistent underlying facial attributes except for the makeup itself.

However, satisfying all three properties simultaneously remains a significant challenge. Real-world makeup datasets Li et al. (2018a); Gu et al. (2019); Nguyen et al. (2021); Dantcheva et al. (2012); Jiang et al. (2020) typically offer high makeup realism ($\mathcal{P}_{makeup}$), but collecting paired images with consistent pose, lighting, and expression is costly and scale-limited. Consequently, they often lack subject diversity and pairwise consistency ($\neg\mathcal{P}_{diversity}$, $\neg\mathcal{P}_{consistency}$). In contrast, synthetic

datasets Gu et al. (2019); Zhang et al. (2024); Lu et al. (2025) enable large-scale pair generation and broader identity coverage, but often fail to produce realistic makeup ($\neg \mathcal{P}_{\text{makeup}}$) or to maintain facial consistency across pairs ($\neg \mathcal{P}_{\text{consistency}}$).

To address these limitations, we develop an improved makeup transfer approach and apply it to transplant real-world makeup styles onto a diverse set of subjects from FFHQ Karras et al. (2019). This strategy leverages authentic cosmetic styles to preserve makeup realism, while substantially enhancing subject diversity to better reflect real-world variability. Due to limited subject diversity ($\mathcal{P}_{\text{diversity}}$) in real-world makeup sources, our method may occasionally fail to perfectly replicate every fine-grained detail of the original makeup. However, rather than pursuing for pixel-perfect makeup reproduction, our goal and key innovation lie in creating a dataset that better fulfills the key criteria of $\mathcal{P}_{\text{makeup}}$, $\mathcal{P}_{\text{diversity}}$, and $\mathcal{P}_{\text{consistency}}$.

This section is organized in three parts: (1) data preparation, (2) method description, and (3) final dataset construction.

## 3.1 DATA PREPARATION

We use existing makeup datasets (MT Li et al. (2018a) and LADN Gu et al. (2019)) as the makeup style source $\mathcal{S}$ to ensure high makeup realism ($\mathcal{P}_{\text{makeup}}$), and employ FFHQ as the target identity set $\mathcal{T}$ to ensure facial diversity ($\mathcal{P}_{\text{diversity}}$). The resulting transferred dataset, denoted as $\mathcal{G}$, is constructed by applying makeup styles from $\mathcal{S}$ onto subjects from $\mathcal{T}$, and is expected to preserve facial consistency ($\mathcal{P}_{\text{consistency}}$) with respect to $\mathcal{T}$.

During training, we exclusively use the makeup dataset $\mathcal{S}$. For each makeup image $I^{\mathcal{S}} \in \mathcal{S}$, we obtain the following data: a facial mask $\mathcal{M}$, detected 2D landmarks $\mathcal{L}$, and a reconstructed 3D face $\mathcal{F}$ via 3DMM fitting Yang et al. (2023). Using the facial mask $\mathcal{M}$, we blend the reconstructed face $\mathcal{F}$ with the background of $I^{\mathcal{S}}$ to produce a reconstructed bare face $\hat{I}_{\text{b}}$. By subtracting this bare face from the original image, we derive the makeup residual $\mathcal{R} = I^{\mathcal{S}} - \hat{I}_{\text{b}}$. To augment the residuals, vertex-wise colors are sampled from $\mathcal{R}$ via the geometry of a reconstructed 3D face $\mathcal{F}$. These sampled colors are then re-rendered onto another reconstructed 3D face $\mathcal{F}^{\mathcal{T}}$, randomly chosen from the target identity set $\mathcal{T}$. This process produces the augmented residual $\tilde{\mathcal{R}}$. Each of the 3,068 makeup images in $\mathcal{S}$ undergoes 100 such augmentations to ensure sufficient diversity.

## 3.2 METHOD

As illustrated in Fig. 3, our method adopts Stable-Makeup Zhang et al. (2024) as the backbone for makeup transfer, which is built upon the pre-trained Stable Diffusion model Rombach et al. (2022). The controllable components are divided into two parts: feature extraction and structural control.

For feature extraction, the Makeup Residual Detail Encoder is designed to extract structure-invariant appearance features from the augmented makeup residual $\tilde{\mathcal{R}}$. This module consists of a frozen CLIP image encoder Radford et al. (2021) and a Makeup Residual Learner, which is based on the channel-attention architecture proposed in FreeUV Yang et al. (2025). This design enables the model to ignore spatial information and selectively capture relevant appearance features. For structural control, structural guidance is provided using ControlNet Zhang

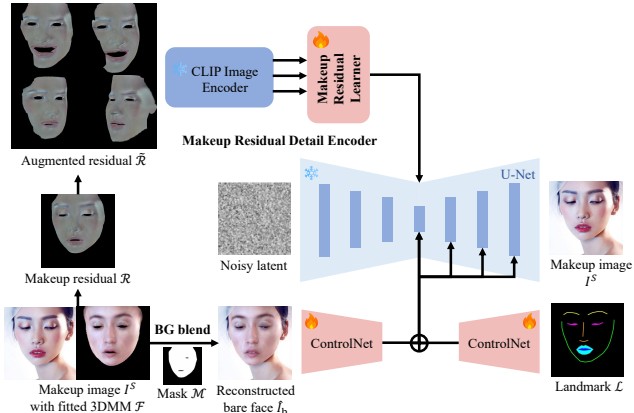

Figure 3: **Overview of the FFHQ-Makeup dataset generation method.** We extract structure-invariant appearance features from augmented makeup residuals, and guide image synthesis with structural priors to ensure facial consistency.

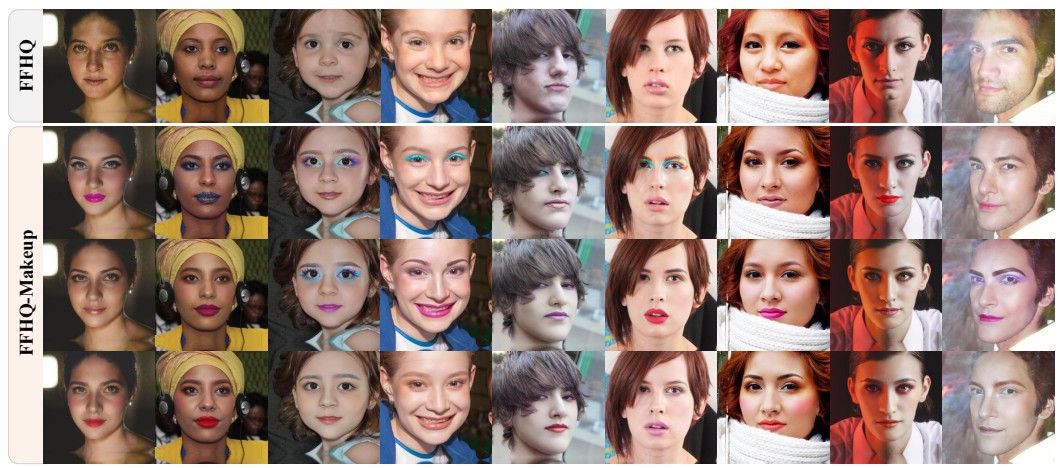

Figure 4: **Examples of FFHQ-Makeup dataset.** Our FFHQ-Makeup dataset inherits the diversity of FFHQ. As shown, it includes multiple bare-makeup pairs examples across different ethnicities, ages, genders, expressions, and cases with occlusions or shadows.

et al. (2023), which leverages both the reconstructed bare face $\hat{I}_b$ and facial landmarks $\mathcal{L}$ to guide the generation process and enhance structural consistency. Notably, unlike Stable-Makeup, our approach removes the reliance on paired bare–makeup data during training. The training setup, including the optimization strategy and hyperparameter configuration, follows that of FreeUV.

### 3.3 DATASET CONSTRUCTION

After training, we apply the trained model for dataset construction. To ensure the quality of the generated dataset, we incorporate careful human inspection and refinement during this phase.

For the makeup source set $\mathcal{S}$ used in appearance extraction, we first remove extreme makeup styles, which are overly rare and may introduce distributional bias. To improve quality, we manually mask out areas in makeup residual $\mathcal{R}$ where facial segmentation fails, particularly in cases with hair overlap or occluded regions. After this cleaning process, a curated subset of 2,257 high-quality makeup residuals is retained.

For the target identity set $\mathcal{T}$ from FFHQ used in structural control, we manually filter out samples exhibiting inaccurate 3DMM fitting or failed facial segmentation. For each identity in $\mathcal{T}$, we randomly select 5 makeup styles from the curated makeup source set $\mathcal{S}$ and perform makeup transfer accordingly. To further refine the results, we apply a mask-guided background blending post-processing: the facial region is taken from the generated makeup image, while the background and clothing regions are preserved from the target identity image, since the generation process often introduces undesirable color shifts in non-facial areas.

After generation, we conduct a group-wise visual inspection to eliminate samples with artifacts or insufficient visual quality. Supplementary details are provided in Appendix 5.1. As a result, the final FFHQ-Makeup dataset comprises 18K unique identities and a total of 90K high-quality images. Examples of the final FFHQ-Makeup dataset are shown in Fig. 2 and 4.

## 4 EVALUATION

We evaluate both the dataset and the data generation method. For dataset-level quantitative evaluation, we compare our FFHQ-Makeup dataset against existing publicly available large-scale synthetic makeup datasets, including LADN-Syn Gu et al. (2019) and BeautyBank Lu et al. (2025). To ensure a fair comparison, we randomly sample 90K images from each dataset to match the scale of our FFHQ-Makeup. For generation method evaluation, we conduct an ablation study to assess different input variants and compare against a baseline that directly uses makeup transfer for data generation.

Table 2: **Visual preference results.** Scores represent the percentage of times each dataset was selected as best in makeup realism ($\mathcal{P}_{\text{makeup}}$) and facial consistency ($\mathcal{P}_{\text{consistency}}$). Our dataset notably outperforms others, especially in maintaining facial consistency.

| | GPT-4o | | | Gemini 2.5 Pro | | | Claude Sonnet 4 | | |
| --- | --- | --- | --- | --- | --- | --- | --- | --- | --- |
| | L-S | BB | **Ours** | L-S | BB | **Ours** | L-S | BB | **Ours** |
| $\mathcal{P}_{\text{makeup}}$ | 0% | 48% | **52%** | 0% | 24% | **76%** | 0% | 46% | **54%** |
| $\mathcal{P}_{\text{consistency}}$ | 0% | 8% | **92%** | 0% | 8% | **92%** | 0% | 18% | **82%** |

L-S = LADN-Syn,    BB = BeautyBank.

In this setting, the evaluation is conducted on a randomly selected subset of 5,000 unfiltered (i.e., not manually cleaned) outputs.

## 4.1 DATASET EVALUATION

As shown in Fig. 1 and 5, LADN-Syn constructs its dataset via a warp-and-paste strategy, which directly overlays makeup regions onto target faces. This naive compositing approach often leads to unrealistic appearances and noticeable artifacts. In contrast, BeautyBank employs a text-to-image generation paradigm. However, due to the inherent ambiguity of natural language prompts, the generated results often exhibit unintended changes in identity and facial expression. Our method, by comparison, produces realistic bare-makeup face pairs while preserving facial structure and identity.

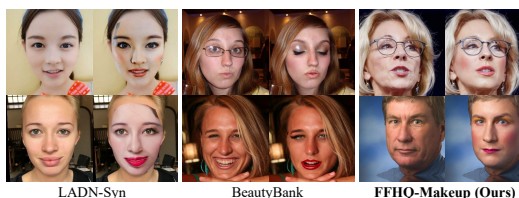

LADN-Syn          BeautyBank          **FFHQ-Makeup (Ours)**

Figure 5: **Qualitative comparison of makeup datasets.** Our FFHQ-Makeup dataset achieves realistic makeup effects while preserving facial structure and identity.

To evaluate dataset quality, we select 50 groups of paired samples and conduct a visual preference study using vision-language models (GPT-4o Hurst et al. (2024), Gemini 2.5 Pro Gemini Team (2025) and Claude Sonnet 4 Anthropic (2025)). The evaluation considers two criteria: makeup realism ($\mathcal{P}_{\text{makeup}}$) and facial consistency ($\mathcal{P}_{\text{consistency}}$), with their definitions. We use the following prompt for the evaluation: "*The following three sets of images are from different makeup datasets. Please select the one you consider the best in terms of Makeup Realism, and Facial Consistency.*" As shown in Tab. 2, our approach achieves the highest scores, especially in facial consistency ($\mathcal{P}_{\text{consistency}}$) across bare–makeup pairs.

## 4.2 ABLATION STUDIES

We compare two ablated training variants to validate the effectiveness of our residual-based representation and augmentation strategy.

The first variant (w/o makeup residual) directly feeds the makeup image $I^S$ into the feature extraction module, instead of using the residual representation $\mathcal{R}$ or $\tilde{\mathcal{R}}$. This setting is similar to Stable-Makeup Zhang et al. (2024), where the source image inherently contains both makeup and identity features. As shown in Fig. 6, this approach leads to entanglement between source identity and makeup, causing identity leakage: the transferred result reflects both makeup style

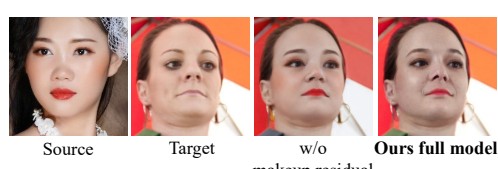

Source          Target          w/o          **Ours full model**
                        makeup residual

Figure 6: **Ablation study on feature extraction without using makeup residual.** Feeding the full makeup image causes identity leakage and entanglement between makeup style and source facial features.

and facial features of the source, thereby compromising identity preservation of the target. In contrast, our residual is derived by subtracting a self-reconstructed face (via 3DMM) from the original, effectively suppressing identity cues and achieving better disentanglement.

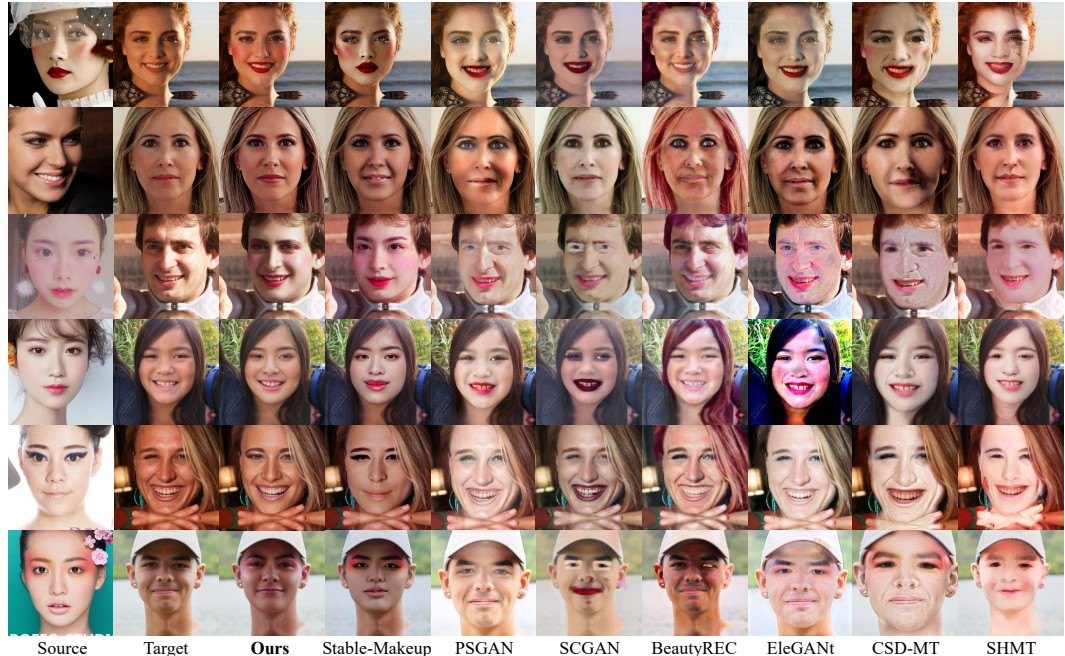

Figure 8: **Comparison of makeup transfer methods for dataset generation.** Our method best preserves the target identity and expression while producing visually plausible makeup. In contrast, other methods often introduce artifacts or alter key facial attributes such as identity and expression.

The second variant (w/o sample and re-rendering) uses the raw residual $\mathcal{R}$ directly without applying the proposed sampling and re-rendering augmentation. As illustrated in Fig. 7, this leads to residual artifacts where subtle structural traits of the source face (e.g., the shape of nostrils) are still present in the output. Our full model mitigates this by re-rendering the residual on diverse FFHQ geometries, encouraging the network to focus purely on makeup-related features and remain invariant to facial structure.

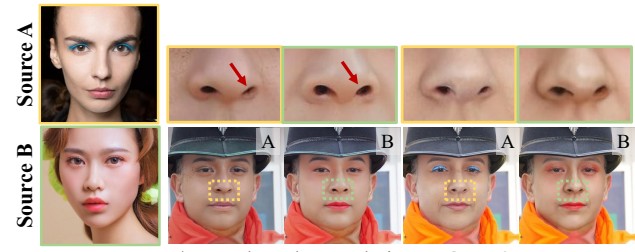

Figure 7: **Ablation study on feature extraction without sampling and re-rendering augmentation.** Direct use of raw residual leads to structural artifacts leaking from the source face into the generated results.

### 4.3 MAKEUP TRANSFER FOR DATASET GENERATION

Our makeup dataset generation method is built upon the Stable-Makeup Zhang et al. (2024) as a baseline, with several improvements introduced to enhance quality and structural control. To validate the effectiveness of our method, we compare it against direct dataset generation using existing makeup transfer methods. Specifically, we evaluate GAN-based methods including PSGAN Jiang et al. (2020), SCGAN Deng et al. (2021), BeautyREC Yan et al. (2023), EleGANt Yang et al. (2022), and CSD-MT Sun et al. (2024b), as well as diffusion-based methods such as SHMT Sun et al. (2024a) and Stable-Makeup Zhang et al. (2024).

As shown in Fig. 8, the GAN-based methods often produce unstable results with significant visual artifacts. While the baseline method Stable-Makeup effectively captures source makeup appearance, it often alters the target's identity and expression, as it relies on paired training data without mechanisms to properly disentangle appearance from structure. In contrast, our method achieves

the best preservation of target identity and expression, while generating a makeup style that closely resembles the source in a perceptually plausible manner.

## 4.4 QUANTITATIVE EVALUATION

We focus the evaluation on the effectiveness of bare–makeup image pairs in maintaining Facial Consistency ($\mathcal{P}_{\text{consistency}}$) using the following metrics. ArcFace Deng et al. (2019) is adopted to measure identity similarity between bare and makeup images, reflecting whether both faces belong to the same person. DINO-I Caron et al. (2021) is employed to assess high-level semantic consistency, such as pose and expression preservation. SSIM is included as

Table 3: **Quantitative comparison of bare–makeup pairs across different dataset and methods.** Both our dataset and method achieve the highest scores in identity similarity and semantic consistency, highlighting their superior ability to preserve facial consistency. **Bolded** values represent the best performance.

|  | Id ↑ | DINO-I ↑ | SSIM ↑ |
|---|---|---|---|
| LADN-Syn | 0.4973 | 0.9163 | **0.9173** |
| BeautyBank | 0.5034 | 0.9008 | 0.8060 |
| **FFHQ-Makeup** | **0.5880** | **0.9561** | 0.8656 |
| Stable-Makeup | 0.5191 | 0.9010 | 0.8431 |
| w/o makeup residual | 0.5346 | 0.9053 | 0.8104 |
| w/o augmentation | 0.5586 | 0.8983 | 0.8126 |
| w/o background blending | 0.5743 | 0.9251 | 0.8206 |
| **Ours full model** | **0.5802** | **0.9478** | **0.8613** |

a low-level structural similarity metric to provide additional insights into pixel-level geometry consistency.

The proposed dataset and its underlying generation method demonstrate superior performance in both identity preservation and semantic structural consistency, validating the effectiveness of our full model in producing high-quality bare–makeup pairs with enhanced facial consistency ($\mathcal{P}_{\text{consistency}}$).

## 5 CONCLUSION AND FUTURE WORK

We presented FFHQ-Makeup, a new large-scale multiple paired bare–makeup dataset with over 90K image pairs, designed to better balance makeup realism, facial diversity, and facial consistency. Each identity is paired with multiple makeup styles, enabling more diverse and robust usage scenarios. To generate high-quality data without relying on paired training data, we introduce a structure-aware diffusion framework that disentangles identity from makeup using 3DMM-guided residuals and facial re-rendering augmentation. Extensive evaluations show that FFHQ-Makeup outperforms existing datasets in both visual quality and facial consistency.

### 5.1 DISCUSSION AND FUTURE WORK

While our method achieves strong makeup realism and facial consistency, the current dataset has several limitations. First, since our approach relies on a relatively small set of makeup reference images, the generated dataset is biased toward common daily makeup styles. As a result, more dramatic or artistic styles are difficult to represent, and some fine-grained makeup details (e.g., subtle eye shadow textures or precise lip contours) are not well preserved. Second, the accuracy of 3DMM fitting and facial segmentation still affects the results, sometimes introducing local artifacts around sensitive regions such as the eyes or teeth, and leaving subtle identity-related cues in the makeup residuals, which hinders full disentanglement. Third, makeup residual strategy is less effective to preserve certain intrinsic identity-related cues, such as moles or facial hair, which can often result in the removal of beards in the generated images.

For future work, we plan to adopt more accurate 3DMM fitting methods (e.g., Pixel3DMM Giebenhain et al. (2025)) to reduce geometric and segmentation errors. Moreover, we plan to expand the variety of makeup sources and adopt more diverse face datasets. We also aim to incorporate automatic evaluation metrics to efficiently scale up the data generation process.

ETHICS STATEMENT

Our work focuses on constructing a synthetic facial makeup dataset. We strictly use publicly available datasets (FFHQ, MT, LADN) that do not contain personally identifiable or sensitive information, and adhere to their respective licenses and usage policies. The generated dataset is intended solely for research in computer vision and beauty-related applications. We are aware that facial datasets may raise concerns regarding privacy, fairness, and potential misuse. We strongly encourage responsible use of our dataset and discourage any applications that may infringe on individual rights or reinforce harmful stereotypes.

REPRODUCIBILITY STATEMENT

We provide detailed descriptions of data preprocessing, model architectures, and evaluation metrics within the paper. Ablation studies are included to clarify the contribution of each component. Upon acceptance, we will release the dataset and code.

LLM USAGE DISCLOSURE

Large language models were used in two ways during this work: (1) to polish the writing style and improve grammar in parts of the manuscript, and (2) as automated evaluators in the visual preference study, where multiple VLMs (Vision-language models) were queried to provide comparative judgments of generated images. The LLMs are did not produce novel scientific content or technical contributions. All statements and claims are verified and edited by the authors. We take full responsibility for all content.

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

## APPENDIX

In this appendix, we provide additional details related to the dataset construction process (Sec. 3.3). Specifically, we illustrate the effect of our mask-guided background blending post-processing, and provide examples of manually filtered-out results during quality control.

### MASK-GUIDED BACKGROUND BLENDING

As shown in Fig. 9, some generated images exhibit undesired background color shifts compared to the target image. To mitigate this issue, we employ a mask-guided blending strategy: the facial region is taken from the generated makeup image, while the background and clothing regions are preserved from the target image. In the blending process, we further apply a morphological erosion with a kernel size of 5 to the facial mask boundary, followed by a Gaussian blur with a kernel size of 15, which helps produce smoother transitions and more natural blending effects. This post-processing step effectively reduces color artifacts outside the facial region.

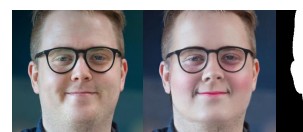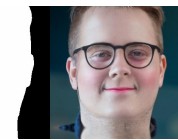

Target    w/o bg blending    Facial mask    **Ours full model**

Figure 9: **Ablation study on mask-guided background blending.**

### FILTERED-OUT SAMPLES

As shown in Fig. 10, we manually discarded samples exhibiting the following issues:

1. The target image itself contains makeup, which contradicts the bare–makeup pairing assumption.

2. Inaccurate 3DMM fitting, particularly around the lips, leading to local mis-alignment and artifacts in the generated images.

3. Generated results with unnatural artifacts. Notably, if one sample in a multi-style group fails, the entire group is discarded to preserve the integrity of the multiple-style pairing.

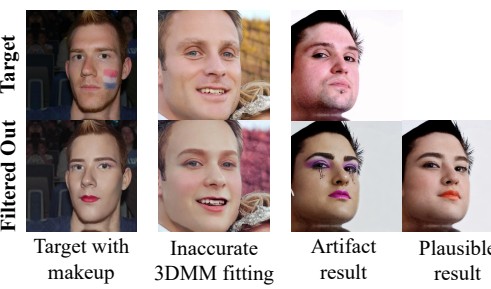

Target with makeup | Inaccurate 3DMM fitting | Artifact result | Plausible result

Figure 10: **Examples of filtered-out results.**

