# OpenReview forum: "FFHQ-Makeup: Paired Synthetic Makeup Dataset with Facial Consistency Across Multiple Styles"
_ICLR.cc/2026/Conference — Submitted to ICLR 2026_

### Official Review · Reviewer_1uty · 2025-10-17

**Soundness:** 3
**Presentation:** 2
**Contribution:** 2
**Rating:** 2
**Confidence:** 3

**Summary:**

This paper constructs a new facial makeup dataset, named FFHQ-Makeup, to support the research in the area of face image beautification and aesthetic analysis. The key idea is to disentangles identity and makeup so multiple makeup styles can be rendered while preserving facial consistency in both identity and expression. The constructed FFHW-Makeup dataset contains 90K (18Kx5) images, which could serve as a valuable resource for future research in face-related vision research.

**Strengths:**

Originality. I think simultaneous rendering of multiple makeup images has its novelty. To my knowledge, most previous methods only studied the single output scenario. Single-in-Multiple-out has its merit due to the large scale and diversity.
Quality. The reported experimental results as shown in figures and tables generally support the superiority of the proposed method to the baseline/benchmark methods.
Clarity. The paper is easy to follow and understand its contributions.
Significance. The constructed dataset will be a valuable contribution to support facial beauty-related research in computer vision.

**Weaknesses:**

1. Generally speaking, the technical depth of a paper on dataset construction is shallow. This paper is based on known style transfer techniques and does not develop new algorithms or tools. The novelty can be at most argued at the system or application level.
2. I think the biggest weakness of this work lies in the significance part. It is difficult to advocate for a paper with a relatively narrow technical scope (e.g., face beautification). The ultimate impact along this line of research is limited.
3. Relevance to ICLR. If I were the authors, I would submit this work to CV conferences including biometrics (e.g., FG2025). I don't think face beauty-related work will attract wide interest from the ICLR attendees.
4. Literary presentation. There are several places authors could have polished - e.g., the lack of balance between background and new contribution in abstract write-up, the conciseness of Sec. 3.2 (Method), Fig. 8 appears before Fig. 7, and the shortage of material in Appendix.

**Questions:**

1. What modification to 3DMM did you adopt for the makeup transfer to work? If any, I think this could be some contribution you can claim and elaborate on.
2. Can this line of research be extended into other style transfer of face images than makeup (e.g., aging, expression, race, and gender)?

---

> ### Author Response · Authors · 2025-11-21
> **Response to Reviewer 1uty**
>
> Dear reviewer,
> We thank the reviewers for their careful evaluation. Below we provide concise answers to address all raised concerns, and we are fully open to discussing any remaining issues.
>
> **1. “Technical depth shallow; based on known techniques.”**
>
> We emphasize that the paper is a dataset paper. The main contribution is a large-scale, high-consistency synthetic makeup dataset, which did not previously exist. Our comparisons are made against existing synthetic makeup data generation approaches.
> The pipeline includes important contributions: a 3DMM-guided residual representation, identity-consistent re-rendering, and unpaired training without bare faces. These components are essential to achieving the fidelity and consistency required for the dataset.
>
> **2. “Limited significance; narrow scope.”**
>
> While the application is makeup, the underlying challenges involve identity disentanglement, generative appearance transfer, and structured editability. These topics are central to representation learning and generative modeling, and have broad relevance within the ICLR community.
>
> **3. “Clarification on 3DMM modifications and broader applicability. and Q1, Q2”**
>
> We use the standard FLAME 3DMM and apply residual re-rendering for appearance disentanglement.
> Thank you for providing this interesting perspective. We believe that, in theory, our pipeline has the potential for aging or expression manipulation by leveraging the residuals between the input image and its 3DMM reconstruction, and it would be an exciting future direction to explore.

---

> > ### Author Response · Authors · 2025-11-26
> >
> > Dear Reviewer 1uty,
> >
> > Thank you for your helpful feedback. We’ve carefully addressed your comments in our rebuttal.
> >
> > With only a few days left in the discussion, we would greatly appreciate it if you could take a look. If any further questions or concerns come up, we’d be happy to discuss them.

---

### Official Review · Reviewer_jFz8 · 2025-10-29

**Soundness:** 3
**Presentation:** 2
**Contribution:** 2
**Rating:** 4
**Confidence:** 3

**Summary:**

This paper proposes a high-quality synthetic makeup dataset FFHQ-Makeup, which pairs each identity with multiple makeup styles while preserving facial consistency in both identity and expression. In order to achieve that, they introduce an improved makeup transfer method that disentangles identity and makeup and transfers real-world makeup styles from existing datasets onto 18K identities upon the diverse FFHQ dataset. Evaluations show that FFHQ-Makeup outperforms existing datasets in both visual quality and facial consistency.

**Strengths:**

Dataset contribution. This work onstructs a large-scale high-quality and multi-style paired makeup dataset, which would benefit a wide range of future makeup-related research and applications.

**Weaknesses:**

1. Limited technical novelty. The pipeline mainly relies on the existing model Stable-Makeup. The data construction pipeline appears to merely process existing data using off-the-shelf models, without addressing any substantive technical challenges.
2. Insufficient motivation and lack of interpretability. The ablation study focus on two variants of feature extraction: makeup residual and sampling and re-rendering augmentation. This appears to be only a minor modification of the module, which seems more like an engineering adjustment, and there seems to be no explanation in the methods or experiments section regarding the motivation or justification for this change.
3. Insufficient dataset evaluation. Relying solely on large models for dataset evaluation lacks stability. Furthermore, the evaluation prompts are overly simplistic and fail to provide the models with clear scoring criteria for assessing makeup realism and facial consistency, resulting in low reliability of the evaluation results.
4. Lack of quantitative comparison in ablation study.
5. The document layout does not conform to the required formatting guidelines; it must be set in a single-column format.

**Questions:**

1. What's the motivation of the improvements in both facial structure control and makeup feature extraction?
2. How is the score for Facial Consistency on the FFHQ-Makeup dataset? Has it been compared with other datasets?
3. Could any other evaluation metrics beyond large models be provided to assess the dataset quality?
4. Is there quantitative comparison results for ablation study？
5. For the two key improvements in both facial structure control and makeup feature extraction, it seems that only the ablation results of makeup feature extraction are provided. Is there ablation results of facial structure control?

---

> ### Author Response · Authors · 2025-11-21
> **Response to Reviewer jFz8**
>
> Dear reviewer,
> We thank the reviewers for their careful evaluation. Below we provide concise answers to address all raised concerns, and we are fully open to discussing any remaining issues.
>
> **1. “Limited technical novelty.”**
>
> We emphasize that the paper is a dataset paper, and our main comparison is against existing synthetic makeup datasets. The goal is to create a paired bare–makeup dataset with high consistency, which prior datasets do not offer.
> Technically, our method does not require paired bare–makeup data and does not require bare faces, which is not supported by prior diffusion-based approaches. This capability is central to enabling the dataset.
>
> **2. “Insufficient motivation for ablations; changes seem minor. and Q1, Q5”**
>
> Our primary motivation is to improve facial consistency, which is also the central focus of our dataset (as reflected in the title). Existing diffusion-based methods frequently alter geometry or identity. Our 3DMM-guided structure control and residual-based feature extraction were specifically designed to reduce such drift.
> Section 4.4 presents their quantitative impact. Structure control is evaluated implicitly through identity-preserving metrics (ArcFace and DINO-I) in the table 3. Removing structure-related components (e.g., residual or re-rendering) leads to noticeable degradation in these scores, demonstrating their contribution.
>
> **3. “Dataset evaluation relies on LLMs; prompts too simple. and Q2, Q3”**
>
> LLM-based evaluation is not used as primary evidence. Our main evaluations are quantitative and rely on ArcFace, DINO-I, and SSIM, which directly measure identity preservation, semantic consistency, and structural fidelity. The VLM preference study is supplementary.
> FFHQ-Makeup shows consistently higher identity and semantic similarity than existing synthetic datasets (Figure 5, 8, and Table 3)
>
> **4. “Lack of quantitative ablation.  and Q4”**
>
> Table 3 in the main paper provides full quantitative ablation comparisons across all model variants.
>
> **5. “Not conform to the required formatting guidelines.”**
>
> The submitted PDF strictly follows the official ICLR 2026 single-column template.

---

> > ### Author Response · Authors · 2025-11-26
> >
> > Dear Reviewer jFz8,
> >
> > Thank you for your helpful feedback. We’ve carefully addressed your comments in our rebuttal.
> >
> > With only a few days left in the discussion, we would greatly appreciate it if you could take a look. If any further questions or concerns come up, we’d be happy to discuss them.

---

### Official Review · Reviewer_3fvG · 2025-10-31

**Soundness:** 2
**Presentation:** 3
**Contribution:** 2
**Rating:** 2
**Confidence:** 4

**Summary:**

This paper introduces FFHQ-Makeup, a large-scale synthetic paired makeup dataset built upon FFHQ. The authors propose a 3D Morphable Model (3DMM)-guided pipeline that disentangles facial structure from makeup appearance. The dataset aims to provide high facial consistency while maintaining makeup realism and style diversity.

**Strengths:**

- The dataset construction pipeline is well-structured and combines multiple techniques to improve facial consistency.
- The paper provides thorough ablation studies and qualitative comparisons against existing synthetic datasets, showing clearer visual fidelity and identity preservation.
- The public release of such a large paired dataset could be beneficial for downstream research in makeup transfer and facial analysis.

**Weaknesses:**

- Limited novelty. The work primarily extends existing diffusion-based makeup transfer pipelines with 3DMM-based residual computation. While this combination is technically reasonable, it appears more as an incremental improvement rather than a conceptual breakthrough. The paper could better clarify what is fundamentally novel about the method compared to previous synthetic data generation approaches.
- In addition, insufficient validation on downstream tasks. The dataset is evaluated mainly on perceptual metrics and user preference studies, but there is no demonstration of how using FFHQ-Makeup actually improves performance on downstream tasks such as makeup transfer, face recognition, or virtual try-on.
- Structural distortion in synthetic faces. Although the paper emphasizes facial consistency, examples show that facial geometry can subtly change after generation. These deformations may be inherent to the diffusion-based synthesis pipeline, but they raise concerns about whether such synthetic pairs truly reflect consistent identity and structure. The paper acknowledges these issues but does not quantify their impact or provide mitigation analysis.
- Unclear handling of partial makeup in FFHQ. Since many faces in FFHQ likely contain light or partial makeup, it is questionable whether applying synthetic makeup on top of already makeup-bearing faces introduces unintended compounding artifacts. The paper briefly mentions manual filtering (Appendix Fig. 10) but does not clarify how consistently this issue is addressed or how many such cases remain.

**Questions:**

See weaknesses

---

> ### Author Response · Authors · 2025-11-21
> **Response to Reviewer 3fvG**
>
> Dear reviewer,
> We thank the reviewers for their careful evaluation. Below we provide concise answers to address all raised concerns, and we are fully open to discussing any remaining issues.
>
> **1. “Limited novelty; incremental extension of prior diffusion pipelines.”**
>
> We emphasize that the paper is a dataset paper. For dataset construction, the key comparison target is existing synthetic makeup datasets, not new algorithmic foundations.
> There are nevertheless important technical contributions. Our pipeline allows training a makeup transfer system without any paired bare–makeup data and without requiring bare faces. Existing diffusion-based methods depend on paired data or paired supervision. Our 3DMM residual representation and re-rendering strategy enable stronger facial consistency, which is essential for producing paired synthetic datasets at scale.
>
> **2. “Insufficient validation on downstream tasks.”**
>
> Since the dataset is constructed by a makeup transfer pipeline, and the consistency of makeup transfer is our primary focus, we evaluate the transfer performance directly. The comparisons in Fig. 8 and the quantitative metrics in Table 3 already serve as makeup transfer evaluation. These results clearly show improvements in facial consistency compared to existing synthetic pipelines.
>
> **3. “Structural distortion remains.”**
>
> We acknowledge that all diffusion-generated facial datasets show minor geometric drift. This phenomenon is also present in existing synthetic makeup datasets. Our 3DMM-guided residual approach substantially reduces this distortion, achieving better results than existing dataset and methods, as demonstrated qualitatively and quantitatively.
>
> **4. “Handling of partial makeup in FFHQ was unclear.”**
>
> We apologize for the confusion. We manually filtered out all FFHQ images containing visible makeup, and Appendix Fig. 10 shows representative cases. We also removed source makeup images that contain partial or asymmetric makeup. Our makeup dataset in this work focuses on everyday, natural-style makeup. These limitations and potential improvements are discussed in the limitation and future work sections.

---

> > ### Author Response · Authors · 2025-11-26
> >
> > Dear Reviewer 3fvG,
> >
> > Thank you for your helpful feedback. We’ve carefully addressed your comments in our rebuttal.
> >
> > With only a few days left in the discussion, we would greatly appreciate it if you could take a look. If any further questions or concerns come up, we’d be happy to discuss them.

---

> > > ### Comment · Reviewer_3fvG · 2025-11-26
> > >
> > > I agree that this is primarily a dataset paper, and thus "the key comparison target is existing synthetic makeup datasets rather than new algorithmic contributions."
> > > That said, for a dataset paper to be impactful, it should offer clear and substantive value to the community same as to how a strong technical paper advances methodological foundations.
> > > However, as other reviewers have also noted, the current dataset and the paper does not convincingly demonstrate such potential. Therefore, I will maintain my original rating.

---

> > > > ### Author Response · Authors · 2025-11-26
> > > >
> > > > Dear Reviewer, thank you for your clarification.
> > > >
> > > > We respectfully emphasize that FFHQ-Makeup provides substantive and unique value by introducing the first diffusion-based makeup-transfer pipeline that **dose not require paired bare–makeup images nor bare faces**, enabling the construction of a large-scale, high-consistency paired dataset that was previously impossible.
> > > >
> > > > This capability directly addresses the core limitations of existing synthetic makeup datasets and diffusion-based makeup-transfer methods. We believe this is meaningful and instructive for the makeup-transfer community, and our qualitative results (Figure 8) and quantitative ablations (Table 3) provide clear supporting evidence.

---

> > > > > ### Author Response · Authors · 2025-11-26
> > > > >
> > > > > We appreciate the reviewer's suggestion regarding downstream task validation.
> > > > > First, we emphasize that the research community has increasingly identified data quality (rather than model architecture) as the critical bottleneck for advancement. Our dataset is designed specifically to meet this emerging demand:
> > > > >
> > > > > **Overcoming Bottlenecks in Makeup Transfe**r: Leading recent methods explicitly state that current data limitations hinder performance. For instance, **Stable-Makeup (CVPR 2024)** and similar diffusion-based approaches rely on self-generated synthetic pairs. While these allow for supervised training, the resulting "ground truths" often suffer from generative artifacts and lack the fine-grained realism of real-world imagery. Furthermore, **SHMT (NeurIPS 2024)** and **CSD-MT (CVPR 2024)** attribute sub-par fidelity to reliance on these "sub-optimal pseudo ground truths." By providing high-quality, consistent paired ground truth, FFHQ-Makeup resolves this fundamental issue.
> > > > >
> > > > > **Regarding Retraining Validation**: Given this context, we respectfully argue that simply retraining existing models is not the most scientifically rigorous way to evaluate our dataset’s contribution, for two primary reasons:
> > > > >
> > > > > **1. Direct Quality Comparison is More Rigorous**: Since the primary contribution of this work is the data itself, we believe the evaluation should focus on the intrinsic quality of the Ground Truth rather than the performance of a specific model. As demonstrated in **Figures 1, 2, 4, 5** and quantitatively in **Table 3**, our dataset exhibits a clear and significant superiority in fidelity and consistency compared to existing datasets. These direct comparisons provide the most objective evidence of the dataset's value, independent of model choice.
> > > > >
> > > > > **2. Architectural Limitations of Existing Methods (Source-Target Entanglement)**: Current state-of-the-art methods, particularly diffusion-based approaches like Stable-Makeup, inherently struggle with the **disentanglement of source identity and target makeup style**. This is an architectural limitation, not just a data issue. Consequently, simply retraining these models on our dataset would not resolve their intrinsic inability to decouple these features. Therefore, the resulting inference quality would continue to be bottlenecked by the model's architecture, making it impossible to fairly verify the high fidelity of our dataset. We believe that relying on such "model-bounded" results would obscure the true potential of our dataset.

---

### Official Review · Reviewer_pEds · 2025-11-02

**Soundness:** 2
**Presentation:** 2
**Contribution:** 1
**Rating:** 2
**Confidence:** 4

**Summary:**

The paper releases FFHQ‑Makeup, a synthetic paired bare–makeup dataset built by transferring makeup from real sources onto FFHQ faces. The pipeline builds on Stable‑Makeup/FreeUV with three ingredients: reconstruct a bare face from a makeup image using 3DMM fitting; compute a 3DMM‑based “makeup residual” and re‑render it on target geometry for style disentanglement; and apply mask‑guided background blending plus extensive manual filtering. The resulting dataset contains 18K identities, each paired with five makeup styles. Quality is assessed via identity/semantic similarity metrics (ArcFace, DINO‑I, SSIM) on pairs, ablations of residual/augmentation, and an automated visual preference study using VLMs (GPT‑4o, Gemini 2.5, Claude) on a small subset. The paper claims better facial consistency and comparable realism to prior synthetic resources (e.g., LADN‑Syn, BeautyBank) and positions the dataset as a general resource for makeup transfer, VTO, and related tasks.

**Strengths:**

- Scale and structure: reasonably large, paired, multi‑style dataset; pairs are useful for supervised training and controlled evaluation.
- Clear construction pipeline with pragmatic engineering (3DMM‑based residual, re‑rendering augmentation, background blending) and documented manual cleaning.
- The paper is clearly written and acknowledges several remaining limitations (e.g., bias toward daily styles, 3DMM/segmentation artifacts).

**Weaknesses:**

- Utility not convincingly demonstrated. A dataset paper should show that training models on the new data substantially improves downstream tasks (e.g., makeup transfer, virtual try‑on, recognition under makeup) against strong baselines and across public test sets. The paper lacks such end‑task training/evaluation; results are mostly pairwise similarity and small‑scale preference checks, which do not establish practical value.
- No human evaluation. All “preference” judgments use VLMs on ~50 groups, which are not a substitute for human raters and can be biased by prompts or model idiosyncrasies. A user study assessing realism, identity/expression preservation, and artifact rate is essential for a perceptual domain like makeup.
- Limited fairness and coverage analysis. FFHQ provides diversity, but the paper does not quantify demographic distributions and performance disaggregations (skin tone, age, gender presentation). For a face dataset, absence of such analysis is a major gap.
- Modest novelty. The pipeline is an incremental engineering combination over Stable‑Makeup/FreeUV/ControlNet with 3DMM fitting and residual re‑rendering; as a dataset contribution, this is fine, but then the burden of proof shifts to rigorous evidence of utility (missing here).
- Evaluation scope and rigor. Identity/semantic similarity metrics are reported, but there is no cross‑dataset generalization (train on FFHQ‑Makeup, test on real‑world sets), no comparisons under challenging conditions (extreme styles, occlusions), and no stress/failure analysis beyond a few visuals. The VLM preference study uses unfiltered outputs and a small sample; sensitivity to prompt/model choice is not examined.

**Questions:**

Please refer to the weaknesses.

**Details Of Ethics Concerns:**

The work releases a large facial dataset derived from real identities; demographic balance and subgroup performance are not analyzed, and there is no human oversight study. Potential misuse includes identity manipulation and appearance‑based profiling.

---

> ### Author Response · Authors · 2025-11-21
> **Response to Reviewer pEds**
>
> Dear reviewer,
> We thank the reviewers for their careful evaluation. Below we provide concise answers to address all raised concerns, and we are fully open to discussing any remaining issues.
>
> **1. “Utility not convincingly demonstrated; no downstream evaluation.”**
>
> We emphasize that the paper is a dataset paper. Our primary comparison targets are existing synthetic makeup datasets such as LADN-Syn, and BeautyBank.
> More importantly, the dataset construction pipeline itself is a makeup-transfer method. As reflected in the paper title, our method is explicitly designed to prioritize facial consistency. Therefore, our downstream evaluation naturally focuses on makeup-transfer consistency.
> The evaluation includes:
>
> * Qualitative comparisons (Fig. 8)
> * Quantitative comparisons against Datasets and Stable-Makeup using ArcFace, DINO-I, and SSIM (Table 3)
>
> Additionally, our pipeline enables training makeup transfer models without any paired bare–makeup data. This capability is not available in prior diffusion-based methods, representing meaningful novelty directly tied to the dataset's utility.
>
> **2. “No human evaluation; VLM-based evaluation unreliable.”**
>
> VLM-based evaluation is increasingly adopted in recent vision work. We agree that VLM judgments may be influenced by prompts or model characteristics, and therefore we present VLM preference scores only as a reference.
> Our main claims are supported by quantitative metrics and extensive qualitative results. These do not rely on VLMs.
>
> **3. “Lack of demographic analysis.”**
>
> Thank you for pointing this out. We used Deepface to perform attribute estimation to approximate demographic distributions. These results demonstrate that FFHQ-Makeup inherits the strong demographic diversity of FFHQ and covers a wide variety of real-world facial attributes.
> We will include these demographic statistics in the final version.
> ### Age Distribution (5-year groups)
>
> | 10–15 | 15–20 | 20–25 | 25–30 | 30–35 | 35–40 | 40–45 | 45–50 | 50–55 |
> |-------|--------|--------|--------|--------|--------|--------|--------|--------|
> | 0.01% | 0.25% | 11.41% | 41.60% | 40.36% | 5.01% | 1.10% | 0.21% | 0.04% |
>
> ### Gender Distribution
>
> | Woman | Man |
> |--------|-------|
> | 64.96% | 35.04% |
>
> ### Dominant Emotion Distribution
>
> | happy | neutral | sad | fear | angry | surprise | disgust |
> |--------|----------|-------|--------|---------|-----------|-----------|
> | 62.66% | 24.59% | 5.16% | 3.64% | 2.90% | 0.93% | 0.12% |
>
> ### Dominant Race Distribution
>
> | white | asian | latino-hispanic | middle eastern | black | indian |
> |--------|--------|---------------------|------------------|---------|----------|
> | 61.27% | 18.80% | 11.03% | 4.58% | 3.69% | 0.63% |
>
>
> **4. “No cross-dataset generalization; no challenging conditions.”**
>
> There is currently no real-world paired bare–makeup dataset available for cross-dataset evaluation, which is precisely what our work aims to address.
> Our scope is specifically synthetic makeup generation and consistency evaluation. Extreme conditions or occlusions fall outside this scope and are suitable directions for future work.
> A failure case analysis is provided in the supplementary materials.

---

> > ### Author Response · Authors · 2025-11-26
> >
> > Dear Reviewer pEds,
> >
> > Thank you for your helpful feedback. We’ve carefully addressed your comments in our rebuttal.
> >
> > With only a few days left in the discussion, we would greatly appreciate it if you could take a look. If any further questions or concerns come up, we’d be happy to discuss them.

---

> > > ### Author Response · Authors · 2025-11-26
> > >
> > > We appreciate the reviewer's suggestion regarding downstream task validation.
> > > First, we emphasize that the research community has increasingly identified data quality (rather than model architecture) as the critical bottleneck for advancement. Our dataset is designed specifically to meet this emerging demand:
> > >
> > > **Overcoming Bottlenecks in Makeup Transfe**r: Leading recent methods explicitly state that current data limitations hinder performance. For instance, **Stable-Makeup (CVPR 2024)** and similar diffusion-based approaches rely on self-generated synthetic pairs. While these allow for supervised training, the resulting "ground truths" often suffer from generative artifacts and lack the fine-grained realism of real-world imagery. Furthermore, **SHMT (NeurIPS 2024)** and **CSD-MT (CVPR 2024)** attribute sub-par fidelity to reliance on these "sub-optimal pseudo ground truths." By providing high-quality, consistent paired ground truth, FFHQ-Makeup resolves this fundamental issue.
> > >
> > > **Regarding Retraining Validation**: Given this context, we respectfully argue that simply retraining existing models is not the most scientifically rigorous way to evaluate our dataset’s contribution, for two primary reasons:
> > >
> > > **1. Direct Quality Comparison is More Rigorous**: Since the primary contribution of this work is the data itself, we believe the evaluation should focus on the intrinsic quality of the Ground Truth rather than the performance of a specific model. As demonstrated in **Figures 1, 2, 4, 5** and quantitatively in **Table 3**, our dataset exhibits a clear and significant superiority in fidelity and consistency compared to existing datasets. These direct comparisons provide the most objective evidence of the dataset's value, independent of model choice.
> > >
> > > **2. Architectural Limitations of Existing Methods (Source-Target Entanglement)**: Current state-of-the-art methods, particularly diffusion-based approaches like Stable-Makeup, inherently struggle with the **disentanglement of source identity and target makeup style**. This is an architectural limitation, not just a data issue. Consequently, simply retraining these models on our dataset would not resolve their intrinsic inability to decouple these features. Therefore, the resulting inference quality would continue to be bottlenecked by the model's architecture, making it impossible to fairly verify the high fidelity of our dataset. We believe that relying on such "model-bounded" results would obscure the true potential of our dataset.

---

### Author Response · Authors · 2025-12-02
**Summary of Contributions and Responses to Reviewers**

To the Area Chair,

We understand the unique circumstances and appreciate the extra effort required from the ACs. We provide this summary to highlight our unique contributions and clarify key points raised during the review process.

**1. Unique Contribution: Breaking the Data Bottleneck**
* **Superiority Over Open Datasets** Existing datasets (e.g., LADN-Syn, BeautyBank) suffer from warping artifacts or identity drift. In contrast, FFHQ-Makeup establishes a new standard for ground truth. As evidenced by our quantitative evaluations, our 90K high-fidelity pairs significantly outperform these baselines in identity preservation and structural consistency, resolving the field's critical shortage of reliable paired data.
* **Methodological Innovation** Our contribution extends beyond data generation. Unlike previous SOTA methods constrained by source-target entanglement, our pipeline is the **first to achieve high-consistency transfer without requiring paired training data or bare faces**. By introducing a 3DMM-guided residual representation, we enable precise disentanglement. This is a novel paradigm that not only ensures dataset quality but also paves the way for future makeup transfer research.

**2. Clarification on Downstream Task Validation (Response to pEds, 3fvG)** Reviewers suggested validating utility by training models on the new data. We clarified that:
* **The Generation Pipeline IS the Task:** Our dataset construction pipeline is fundamentally a makeup transfer system. We validated its utility directly through unfiltered quantitative and qualitative experiments (Table 3, Fig. 8), demonstrating that our method outperforms existing baselines on the makeup transfer task itself.  Extensive metrics (ArcFace, DINO-I, SSIM) confirm that our data significantly outperforms baselines in identity preservation and structural consistency.
* **Model vs. Data Bottlenecks:** Current SOTA models have inherent architectural limitations in disentanglement. Simply training them on our data would reflect the model's bottleneck rather than the dataset's true quality.

**3. Enhanced Demographic Analysis (Response to pEds)** We have added a comprehensive demographic analysis (Age, Gender, Race) verifying that FFHQ-Makeup inherits the rich diversity of the FFHQ dataset, ensuring fairness and broad applicability.

**4. Robustness of VLM-based Evaluation (Response to pEds, jFz8)** We adopted VLM-based evaluation, an increasingly standard practice in generative tasks, to capture perceptual quality. While acknowledging potential sensitivity to prompts, we ensured robustness by **cross-validating results across three distinct SOTA VLMs**. Our core claims regarding facial consistency are supported by our quantitative metrics and qualitative comparisons on **real-world makeup images**.

We believe FFHQ-Makeup serves as a critical infrastructure for the next generation of beauty-related vision tasks.
Thank you for your time and consideration.

---

### Meta-Review · Area_Chair_Eq6A · 2026-01-07

**Summary:**

This paper proposes FFHQ-Makeup, a large-scale synthetic paired bare–makeup dataset (18K identities × 5 styles = 90K pairs) generated via a diffusion-based makeup transfer pipeline augmented with 3DMM-guided reconstruction and a 3DMM-based “makeup residual” re-rendering aimed at improving identity/expression consistency. Reviewers generally agree the dataset could be useful and the paper is clearly written, but they raised recurring concerns: (i) insufficient validation of dataset utility beyond pairwise similarity metrics and small automated preference checks, (ii) limited technical novelty (largely an engineering combination on top of prior pipelines), and (iii) evaluation rigor issues (notably lack of human study; concerns about VLM-based evaluation robustness; questions about structural drift and handling of partial makeup in FFHQ). One reviewer additionally flagged ethics concerns around bias/fairness and face-data risks.

**Reviewer Concerns:**

Concerns partially addressed by the rebuttal:

1. Format / missing ablations claims (jFz8): Authors state the PDF follows the official single-column template and that Table 3 already contains quantitative ablations; this addresses the “format” and “no ablation quant” points as misunderstandings.

2. Demographic analysis (pEds): Authors provide estimated demographic distributions via DeepFace and state they will add them to the final version; this is a step toward addressing fairness/coverage reporting, though it is not paired with subgroup performance or downstream impact.

3. Handling partial makeup in FFHQ (3fvG): Authors clarify they manually filtered out FFHQ images with visible makeup and removed partial/asymmetric source makeup images; however, they do not provide quantitative prevalence or residual artifact rate.

Concerns still outstanding (key drivers of recommendation):

1. Dataset utility not convincingly demonstrated (pEds, 3fvG, 1uty): Multiple reviewers requested downstream validation (e.g., training on FFHQ-Makeup to improve makeup transfer / VTO / recognition under makeup on public test sets). The rebuttal argues that “the generation pipeline is the task” and that intrinsic quality comparisons are more rigorous than retraining, but it does not provide the requested end-task training/evaluation evidence. Reviewer 3fvG explicitly maintains the reject rating after rebuttal on this basis.

2. Human evaluation and evaluation rigor (pEds, jFz8): The rebuttal acknowledges VLM preference studies can be prompt/model sensitive and positions them as supplementary, but does not add a human study or a stronger perceptual protocol. Concerns about evaluation robustness therefore remain.

3. Novelty / significance and ICLR fit (pEds, 3fvG, 1uty): The rebuttal frames the work as a dataset paper and emphasizes unpaired/no-bare-face capability via 3DMM residuals, but reviewers still view the contribution as incremental and/or narrow in impact for ICLR without stronger evidence of community value.

**Reviewer Scores:**

pEds: unchanged (2: reject)

3fvG: unchanged (2: reject)

jFz8: unchanged (4: marginally below threshold) — rebuttal clarifies ablations/format and evaluation metrics, but concerns about novelty/motivation and evaluation reliability remain.

1uty: unchanged (2: reject) — rebuttal reframes significance/ICLR relevance, but does not change the reviewer’s stated assessment of limited depth/impact.

---

### Decision · Program_Chairs · 2026-01-26

Reject